# Adaptive and Biological Responses of Buffalo Granulosa Cells Exposed to Heat Stress under In Vitro Condition

**DOI:** 10.3390/ani11030794

**Published:** 2021-03-12

**Authors:** Marwa S. Faheem, Nasser Ghanem, Ahmed Gad, Radek Procházka, Sherif M. Dessouki

**Affiliations:** 1Department of Animal Production, Faculty of Agriculture, Cairo University, 12613 Giza, Egypt; marwasf@agr.cu.edu.eg (M.S.F.); nassergo@agr.cu.edu.eg (N.G.); sherifd2002@agr.cu.edu.eg (S.M.D.); 2Cairo University Research Park (CURP), Faculty of Agriculture, Cairo University, 12613 Giza, Egypt; 3Laboratory of Developmental Biology, Institute of Animal Physiology and Genetics of the Czech Academy of Sciences, 27721 Liběchov, Czech Republic; Prochazka@iapg.cas.cz

**Keywords:** granulosa cell, steroidogenesis, heat stress, gene expression, miRNAs, buffalo

## Abstract

**Simple Summary:**

The pertinent consequences of global warming substantially affect both animal productivity and fertility. Noteworthy, tropical and subtropical animal breeds are productively suited to hot climate conditions. Therefore studying the physiological changes accompanying high temperature, especially in tropically adapted species such as buffalo, will help in understanding the mechanisms that the animal use to accomplish the necessary functions efficiently. Concerning fertility-related activity, granulosa cells are important for the regulation of ovarian function and the completion of oocyte maturation. In this study, the buffalo granulosa cells were examined for their viability, physiological and molecular responses under in vitro heat stress conditions. Buffalo granulosa cells displayed different adaptive responses, at the physiological and molecular levels, to the different heat stress conditions. At 40.5 °C, granulosa cells exhibited a functional persistence compared to the control and other heat-treated groups. These results will provide insights into ways that tropically adapted breeds may be able to maintain better reproductive function when exposed to heat stress compared to temperate breeds.

**Abstract:**

The steroidogenesis capacity and adaptive response of follicular granulosa cells (GCs) to heat stress were assessed together with the underlying regulating molecular mechanisms in Egyptian buffalo. In vitro cultured GCs were exposed to heat stress treatments at 39.5, 40.5, or 41.5 °C for the final 24 h of the culture period (7 days), while the control group was kept under normal conditions (37 °C). Comparable viability was observed between the control and heat-treated GCs at 39.5 and 40.5 °C. A higher release of E2, P4 and IGF-1 was observed in the 40.5 °C group compared with the 39.5 or 41.5 °C groups. The total antioxidant capacity was higher in response to heat stress at 39.5 °C. At 40.5 °C, a significant upregulation pattern was found in the expression of the stress resistance transcripts (*SOD2* and *NFE2L2*) and of *CPT2*. The relative abundance of *ATP5F1A* was significantly downregulated for all heat-treated groups compared to the control, while *TNFα* was downregulated in GCs at 39.5 °C. Expression analyses of stress-related miRNAs (miR-1246, miR-181a and miR-27b) exhibited a significant downregulation in the 40.5 °C group compared to the control, whereas miR-708 was upregulated in the 39.5 and 40.5 °C groups. In conclusion, buffalo GCs exhibited different adaptive responses, to the different heat stress conditions. The integration mechanism between the molecular and secretory actions of the GCs cultured at 40.5 °C might provide possible insights into the biological mechanism through which buffalo GCs react to heat stress.

## 1. Introduction

Granulosa cells (GCs) play a critical role during the growth and development of mammalian ovarian follicles. The secretory and nutritive supportive properties, along with the synergistic physical and functional associations, are the most important distinguishing features of their role. These characteristics are not constant, and are in a continuous dynamic with both the developmental stage of the growing ovarian follicle [1,2,3] and the estrus cycle of the animal [4]. Based on this dynamic, the oocyte can successfully realize its cellular and functional potential, resulting in a competent oocyte for the subsequent developmental process. This mutual kinetic responsiveness, inside the follicular compartment, is mediated via gap junction communication in the form of transzonal projections (TZPs), which originate from GCs [5]. This in turn enables GCs to coordinate nuclear and cytoplasmic maturation [6] and regulate the transcriptional activity of the oocyte [7]. Furthermore, GCs as a self-functioned structure is concerned mainly with steroidogenesis in order to regulate the ovarian function in a responsive action with animal reproductive status. Different environmental stressors, e.g., heat stress, demonstrated dramatic influences on the GC estrogenic activity of heat-stressed ovarian follicles, which triggered their susceptibility to apoptosis [8,9]. Accordingly, the variation in GC behavior towards steroidogenesis in response to a certain stress could be effectively monitored under defined conditions in vitro.

The cellular response to heat stress involved several functions and pathways that are regulated at the transcriptional and posttranscriptional levels. MicroRNAs (miRNAs), small non-coding RNA molecules (19–25 nucleotides long), are key global regulators of gene expression at the posttranscriptional level. Under different stress conditions, miRNAs could partly mediate the cellular response by altering the expression of target genes and/or modulating the activity of miRNA-protein complexes [10]. A recent study by Shandilya et al. [11] indicated that cellular and molecular responses to heat stress, including gene and miRNA expression, are correlated with species-specific adaptive capacity. Therefore, understanding the regulatory mechanisms of the cellular tolerance and response to heat stress in different mammalian species will provide more insights into the diverse range of adaptation mechanisms of these species under different environmental conditions. Among the commonly used models of ovarian function, cattle, a monovular species, display patterns of follicular development that are very similar to those in humans. In that context, buffalo is an appropriate animal model for understanding the underlying mechanism of adaptability to hot climate conditions in terms of ovarian functions. The core body temperature of this animal ranges from 38.5 °C to 41.5 °C and the rectal temperature might raise up to 41 °C in response to hot weather [12]. Therefore, the objective of this study was to determine the adaptive and biological responses of buffalo GCs towards steroidogenesis under heat stress conditions and to determine the underlying regulating mechanisms at the molecular level.

## 2. Materials and Methods

All cell culture reagents and media were obtained from Sigma–Aldrich Chemical Company (St. Louis, MO, USA) unless stated otherwise. There was no direct contact with the animals. All experiments were performed using materials obtained from a slaughterhouse.

### 2.1. Collection of Ovaries and Granulosa Cells

Ovaries from healthy and cyclic female buffaloes (of age 4–8 years) were obtained from a local abattoir post their slaughter. Then, ovaries (20–30 ovaries/collection for three different collection days) were pooled in a thermos containing warm saline solution (0.9% NaCl) at 34–37 °C and brought within 2 h to the laboratory. After removal of the adhering tissues, ovaries were subjected to three washing steps with different warm saline solutions followed by 30 s of rinsing with ethanol (70%) and kept in a water bath at 37 °C after the final washing (twice) in fresh warm saline solution.

The follicular fluid of subordinate antral follicles (with a diameter of 2–8 mm) containing the GCs and cumulus-oocyte complexes (COCs) was aspirated using an 18-gauge needle. The follicular fluid was pooled for oocyte sedimentation, and the precipitate was checked for COCs removal. The remaining cell clots were collected and allocated into a 15 mL sterilized falcon tube for 10 min of centrifugation at 177× *g*. After discarding the supernatant, 0.25% trypsin-EDTA was used for 5 min to dissociate the clumps of GCs in TCM-199 HEPES medium (TCM-199 containing 1.5% streptomycin and penicillin G potassium) with gentle pipetting. Then followed by 5 min of centrifugation at 277× *g*. After discarding the supernatant, 1 mL of prewarmed complete culture medium (TCM-199 HEPES supplemented with 10% FBS and 1.5% antibiotics) was added to resuspended GCs pellet. Granulosa cell viability and concentration were determined using the trypan blue exclusion method (0.4%) according to Baufeld and Vanselow [13].

### 2.2. Culture of Granulosa Cell under Heat Stress Conditions

Granulosa cells (1 × 10^6^ cells/mL) were cultured in a tissue culture flask (Nunclon Delta, Thermo Fisher, Roskilde, Denmark) and incubated in a humidified air under 5% CO_2_ at 37 °C in a complete culture medium (TCM-199 HEPES supplemented with 10% FBS and 1.5% antibiotics) for two days until they reached a confluence of 80–90%. The medium was replaced with a fresh one and GCs were then cultured for 6 days in a humidified air under 5% CO_2_ at 37 °C, with medium replacement every two days. On day 7, GCs were assessed microscopically for their viability and homogeneity and then cultured for 24 h under different heat stress conditions (39.5, 40.5, or 41.5 °C) or under normal condition (37 °C) as a control group. At the end of the treatment, cultured GCs and their corresponding conditioned media were harvested separately from post heat-treated GCs and control groups for further assessment. The spent media were kept at −80 °C for hormonal and enzyme assays, while GCs were detached from the culture flasks using trypsin-EDTA (0.25%), reassessed for viability, and kept at −80 °C for gene and microRNA expression analyses.

### 2.3. Determination of Estradiol (E2), Progesterone (P4), and Insulin-Like Growth Factor 1 (IGF-1) by ELISA

Hormonal levels of E_2_, P_4_, and IGF-1 were assessed in the spent conditioned media harvested from control and heat-treated groups by enzyme-linked immunosorbent assay (ELISA) using kits designed for bovines (SinoGeneClon Biotech CO., Ltd.; HangZhou, China) according to the manufacturer’s guidelines. Three biological samples were used to perform each analysis and the optical density (OD) was detected using a Bio Tek Instrument, Inc., San Diego, CA, USA, ELX808 (USA) reader at 450 nm wavelength. The sensitivity of the E_2_, P_4_, and IGF-1 assays was 40 pg/mL, 8.57 pg/mL, and 1 ng/mL, respectively.

### 2.4. Determination of Total Antioxidant Capacity (TAC) and Superoxide Dismutase Enzyme (SOD) Levels

For enzymes assay, a fraction of the spent conditioned media from the same three biological samples used for hormonal analyses were subjected to the colorimetric method to assess the TAC and SOD levels at 505 nm wavelength by a spectrophotometer (Sunostk SBA 733 plus, Bio diagnostic Egypt, Giza Governorate, Egypt).

### 2.5. Gene Expression Analysis

#### 2.5.1. RNA Isolation and cDNA Synthesis

Three biological aliquots of each group of granulosa cells collected from three different trials were used to isolate RNA using Thermo Scientific GeneJET RNA extraction Kit (Vilnius, Lithuania) according to the manufacturer’s guidelines.

For DNA digestion, 1 μL of DNases and 1 μL of MgCl_2_ buffer (Thermo Scientific, Waltham, MA, USA) was added to the isolated RNA samples and incubated at 37 °C for 30 min in a PCR instrument (Thermo Scientific, Waltham, MA, USA). The PCR reaction was deactivated by adding 1 μL of EDTA at 65 °C for 10 min in a PCR instrument. The purity and concentration of the isolated total RNA were checked using a NanoDrop 2000C (Thermofisher Scientific, Wilmington, DE, USA). Samples with 260/280 ratio ≥ 1.8 were used for further analysis. All RNA samples were adjusted to the same concentration (1 μg total RNA) by adding different volumes of RNAse-free water. The cDNA synthesis of all samples was performed using a RevertAid cDNA synthesis kit (Thermofisher Scientific, Vilnius, Lithuania) according to the manufacturer’s guidelines.

#### 2.5.2. Quantitative Real-Time PCR (qRT-PCR)

Primers for the selected genes (Table 1) were designed using the free software Primer3web (http://primer3.wi.mit.edu//; accessed on 15 January 2020) according to available sequences in the GenBank database (www.ncbi.nlm.nih.gov; accessed on 15 January 2020).

Real-time PCR reaction was conducted using Maxima SYBR Green/ROX qPCR Master Mix (Thermofisher Scientific, Vilnius, Lithuania) in a StepOnePlus instrument (Applied Biosystems, Foster City, CA, USA) in a reaction mixture (20 μL) consisting of 10 μL of SYBR green with ROX, 7.8 μL of nuclease-free water, 0.2 μL of specific-reverse primer, 0.2 μL of specific-forward primer, and 2 μL of the cDNA sample. The reaction conditions were as follows: 50 °C for 2 min, initial denaturation at 95 °C for 10 min, then 40 PCR cycles consisting of 15 s for denaturation at 95 °C, 1 min for annealing at 60 °C, and 30 s for extension at 72 °C; while 5 min for final extension at 72 °C. Data of gene expression were analyzed using the delta–delta Ct method and reported as relative transcript abundance after normalization to the housekeeping gene Glyceraldehyde 3-phosphate dehydrogenase (*GAPDH*) according to Khan et al. 2020 [14] and Gebremedhn et al. [15].

### 2.6. Expression Analysis of microRNAs

#### 2.6.1. Total RNA Isolation and Quality Control

Total RNA enriched with miRNA was isolated from GCs of the different groups (heat-treated GCs and control groups) using a mirVana™ miRNA Isolation Kit (Life Technologies, Carlsbad, CA, USA) according to the manufacturer’s instructions. RNA quality and quantity were assessed with an Agilent 2100 Bioanalyzer (Agilent Technologies, Santa Clara, CA, USA), and a NanoDrop 8000 spectrophotometer (NanoDrop Technologies, Wilmington, DE, USA), respectively.

#### 2.6.2. Quantification Analysis of Selected miRNAs Using Droplet Digital PCR (ddPCR)

A total of 10 ng RNA per reaction was reverse transcribed using a TaqMan MicroRNA Reverse Transcription Kit (Thermo Fisher Scientific, Waltham, MA, USA) according to the manufacturer’s instructions. TaqMan miRNA assays (Applied Biosystems, Foster City, CA, USA) were used to quantify the copy numbers of the selected miRNAs in granulosa cell samples (in triplicates) on a QuantaLife QX200 droplet digital PCR (ddPCR) system (Bio-Rad Inc., Feldkirchen, Germany) as previously described [15]. The absolute copy number of each miRNA was determined using QuantaSoft software (version 1.7). The expression of miRNAs was normalized to miR-361 and miR-99a-5p, as both exhibited the most stable expression in bovine GCs under normal and heat stress conditions [15].

### 2.7. Statistical Analysis

Data were checked for model assumptions and we found no significant heterogeneity of variance between groups according to LEVENE/BARTLETT tests. In addition, we tested the distribution of data and we found all observations were normally distributed according to the Shapiro–Wilk test. The viability, hormonal, and enzymatic data of the cultured granulosa cell groups were analyzed using one-way analysis of variance (ANOVA) and expressed as mean ± standard error of the mean (SEM). The program IBM SPSS Statistics 22 (SPSS Inc., Chicago, IL, USA) was used. Gene and miRNA expression data were analyzed using the SAS GLM procedure. Duncan’s multiple range test was used to detect differences between the means. The statistical significance level was defined at *p* < 0.05.

## 3. Results

### 3.1. Granulosa Cell Viability under Heat Stress Conditions

GC viability was estimated in the different heat-treated groups using the trypan blue exclusion method. Results showed that GCs exposed to 41.5 °C exhibited a significant reduction in their viability (82.6% ± 2.1%) compared to the control (93.5% ± 2.0%) and the 40.5 °C group (90.3% ± 2.0%) but not to the 39.5 °C group (86.9% ± 3.1%) (Figure 1). Moreover, the viability of GCs exposed to either 39.5 or 40.5 °C was comparable with the control group (37 °C).

### 3.2. Steroidogenesis and Enzymatic Activities

The concentration level of E_2_ and P_4_ hormones and IGF-1 was analyzed in the spent culture media to determine the effect of heat stress on the functionality of the treated GCs. The three hormonal levels exhibited the same pattern among the heat-treated when compared to the control group. As shown in Figure 2A–C, the concentration of each hormone was significantly decreased in the 39.5 °C and 41.5 °C groups compared to the 40.5 °C and control group, which showed no significant differences from each other.

On the other hand, the impact of heat stress on the antioxidant capacity of heat-treated GCs was estimated by measuring the TAC and SOD2 activity in the spent conditioned media. The TAC level was significantly higher in the 39.5 °C group compared to the control and 40.5 °C groups (Figure 2D), whereas the SOD2 activity exhibited no significant differences between all the groups in the study (Figure 2E).

### 3.3. Gene Expression Patterns in Heat-Treated GCs

The qRT-PCR analysis was performed on the heat-treated GCs to measure the relative expression of genes known to be involved in the cellular defense mechanisms against stress and metabolic activities. The relative abundance of *SOD2* and *NFE2L2* (also known as *NRF2*) genes were significantly upregulated (more than 15- and 6-fold, respectively) in the 40.5 °C group compared to the other groups (Figure 3). Moreover, the expression of *SOD2* was significantly upregulated in the 41.5 °C group compared to the control and 39.5 °C groups (Figure 3).

Similarly, the transcriptional abundance of *CPT2*, a gene that encodes metabolic activity, was significantly upregulated more than 25-fold in the 40.5 °C group compared to all other groups. Although the transcript abundance of this gene was downregulated at 39.5 and 41.5 °C, no significant differences were observed when compared to the control group (Figure 3). The expression of the tumor necrosis factor-alpha (*TNFα*) gene revealed a significant downregulation in the 39.5 °C group compared to the other GC groups (Figure 3). Furthermore, the mRNA abundance of *ATP5F1A* transcript was significantly downregulated in all heat-treated GCs compared to the control group.

### 3.4. MiRNAs Expression Patterns in Heat-Treated GCs

A group of miRNAs previously known to be correlated with stress factors was quantified in the different GCs groups. Expression analysis showed that miR-1246 was significantly downregulated more than 20-fold in the 40.5 °C compared to the control group. The same miRNA was also downregulated in the 39.5 °C group more than 2-fold compared to the control group. The same expression pattern was noticed for miR-181a and miR-27b, in that both miRNAs were significantly downregulated in the 40.5 °C group compared to the control group. On the other hand, miR-708 was upregulated in the 39.5 °C and 40.5 °C groups compared to the control (Figure 4).

## 4. Discussion

The differences in the physiological features and reproductive performance of buffalo species in comparison to cattle reveal that the buffalo is an animal that is specifically adapted to subtropical regions. This distinct adaptive behavior successfully serves to accomplish physiological functions. Therefore, the impact of induced heat stress on the physiological and molecular responses of in vitro cultured buffalo GCs was assessed in this study.

Interestingly, the viability of GCs and their secretory functions related to steroids and growth factor were compatible in the heat-treated GCs when cultured at 40.5 °C in comparison to their counterpart cultured under normal temperature conditions. This observation was consistent with the finding of Zeebaree et al. [16] regarding steroidogenic capacity for heat-treated GCs in bovine. The sustainable release of steroids under 40.5 °C, as revealed by our study, could act as a self-promoter to strengthen the functionality of their secreted cells, as estradiol has a protective effect against induced apoptotic factors [17]. In addition, it might reinforce the expression of genes that regulate steroidogenesis, as revealed by the expression of StAR, Cyp11a1, and Cyp19a1 in mice [18] and *HSD3B1* in cattle [16] being normal at 40.5 °C. The symmetry in IGF-1 activity with the synthesis of steroid hormones of GCs sustained at 40.5 °C might explain the stimulation effect of IGF-1 towards steroidogenesis besides its repression action in GC apoptosis [19,20]. In a retrospective study, the upregulation of steroidogenic-regulating genes in addition to promoting estradiol release were the underlying regulating action of IGF-1 in the cultured GCs in bovine [21]. It has been also acknowledged that IGF-1 mechanisms could act through promoting StAR expression, which is an essential steroid-regulating gene [22] or regulating proapoptotic Bcl-2 factor and related proteins [23,24]. Of relevance, the expression of Bcl-2 was stable and not involved in the apoptotic mechanism of GCs cultured under the heat-induced conditions at 40.5 °C in mice [18] and 43 °C in sheep [18,25].

The stable pattern of SOD2 activity, being an antioxidant enzyme, among the heat-treated GC groups could explain the persistence of GC viability under heat stress conditions. It has been previously indicated that a positive correlation between the antioxidative enzyme activity and the progression of follicular atresia is indicated by GC apoptosis [26,27]. Besides this, the inhibition of apoptosis in follicular cells is a further functional mechanism of SOD [28]. While we found this association to be irrespective of the steroidogenesis performance of heat-treated GCs, at 40.5 °C, the active steroid synthesis from GCs might accelerate the generation of reactive oxygen species (ROS) [14], in turn causing a clear increase in *NFE2L2* and *SOD2* gene expression. These increases in stress resistance transcripts as shown in the current study might be considered as a defense mechanism against high levels of superoxide radical-associated metabolic activity in response to active steroid secretion [28,29,30,31]. Thus excessive ROS production was counteracted by activating scavenging machinery through the (*NFE2L2*) NRF2-mediated oxidative stress response mechanism and its downstream antioxidant genes in bovine GCs [32] and preimplantation embryos [29]. However, bovine GCs were unable to exhibit sufficient cellular protection at a prolonged exposure to 40 °C despite the rise in heat shock response protein as a defense mechanism [14]. Species-specific characteristics of buffalo might give them the ability to adapt to a higher temperature, including long-term exposure, compared to temperate cattle breeds, and the activation of oxidative stress response might be the underlying regulating mechanism [33]. This theory was confirmed through the acceleration of fatty acid (FA) cellular metabolism detected as a dramatic upregulation of carnitine palmitoyltransferase (*CPT2*) transcripts at 40.5 °C, the key cytokine for free FA transport into mitochondria to undergo FA oxidation [34]. Thus becoming an intrinsic source of energy required for active steroid secretion, while among the heat-treated GC groups, mitochondria homeostasis might remain via the constant expression of the *ATP5F1A* gene [35] and SOD2 enzyme stability [36] to maintain GC viability under heat stress conditions instead of participating in steroidogenesis. The downregulation tendency of stress resistance and metabolic activity-related transcripts for the GCs cultured at 39.5 °C might act synergistically, thus resulting in compromised GCs functionality, which was concomitant with the significant downregulation of the *TNF-α* gene in this group when compared to the other studied groups.

The integration mechanism between the molecular and secretory actions of the GCs cultured under 40.5 °C might lead to suppressing the oxidative stress that is likely responsible for the proper emergence of antioxidant activity. Furthermore, this finding is possibly affined to the antioxidant properties of E2 [37]. In agreement with our results, a correlation between higher antioxidant capacity and steroidogenesis deficiencies has been reported in humans [38].

In the current study, we found an uncommon pattern of cellular response to different heat stress intensities. This pattern appeared clearly in most of the analyses including the hormonal activities, antioxidant capacity, gene, and miRNA expressions. In which the GC’s responses to heat stress were similar at 39.5 °C and 41.5 °C but different at 40.5 °C. The GCs exposed to 40.5 °C showed an ability to regulate gene and miRNA expressions and kept the steroidogenic activity as the control group compared to other heat-treated groups. A similar pattern has been reported in bovine GCs exposed to different heat stress intensities specifically in ROS levels and apoptotic rate [14]. The authors suggested that this might be due to the ability of the cells to activate their antioxidant system and regulating gene expression at specific heat intensity. Not only the heat intensity as a mean of different temperatures but also the time of exposure at the same temperature. Bovine GCs exposed to heat stress at 41 °C for 48 h exhibited an enhanced resistance pattern when compared to 24 h in terms of cell proliferation, ROS accumulation, and expression of stress-related genes [39]. These results together with our findings could suggest that cellular response to heat stress may be modulated by the stress intensity and could be species-specific. Moreover, the sustainability of cellular homeostasis under stress is regulated by several mechanisms that are correlated to the stress conditions and genotypes [40]. However, more studies including species comparisons are still needed to elucidate this point.

In this study, we analyzed the expression profiles of four different stress-related miRNAs in GCs under different heat stress conditions. Interestingly, miR-1246 exhibited a severe reduction in expression in the GCs cultured at 40.5 °C. In two independent studies, miR-1246 was reported to be highly expressed in the blood plasma [41] and serum [42] of cows exposed to heat stress. Moreover, extracellular vesicles (EVs) released from heat-stressed GCs in the culture media were found to be enriched in miR-1246 compared to EVs from the unstressed cells [15]. On the other hand, cattle and buffalo fibroblast cells exhibited a reduction in miR-1246 expression in the first few hours after the recovery from exposure to heat stress [11]. Additionally, the inhibition of miR-1246 improves the viability of retinal pigment epithelial cells after exposure to oxidative stress [43]. In silico pathway analysis revealed that genes related to the oxidative stress response pathway are among the target genes regulated by miR-1246 [44]. These results together with our finding suggested that in response to stress conditions, the intracellular reduction of miR-1246 expression could be a cellular protective mechanism to regulate stress-related genes and maintain cell viability. This reduction could be through the release of EVs into the circulation system or the culture media. However, the exact function of the released miR-1246 in the target recipient cells under stress conditions needs more investigation. In correlation with GCs functions, predicted target genes of miR-1246 were known to play a role in progesterone biosynthesis regulation [45,46]. Moreover, a significant downregulation of miR-1246 expression was reported in large compared to small bovine follicles [47]. This indicates a negative regulation mechanism of miR-1246 on GC secretory functions, and could explain the ability of the GCs cultured at 40.5 °C to maintain their steroid secretory function.

Furthermore, we analyzed the expression pattern of miR-181a and miR-27b in the different GC groups. Both miRNAs are known to be involved in heat stress and immune responses in cattle and buffalo [11,42]. In this study, miR-181a and miR-27b exhibited a similar pattern to miR-1246, with a more than twofold reduction in GCs cultured at 40.5 °C. Previously, miR-181a was reported to be a regulator of cell proliferation and apoptosis [48]. In humans, the induction of apoptosis and ROS production was suppressed by the inhibition of miR-181a expression in chondrocyte cells [49]. In another study, Chen et al. [50] reported that the inhibition of miR-181a in bovine peripheral blood mononuclear cells can reduce heat stress damage through the upregulation of antioxidant-related genes and downregulation of apoptotic genes. Additionally, miR-27b has been proved to target Nrf2 mRNA and subsequently reduce the expression of its downstream antioxidant genes [51]. In agreement with these findings, our results showed a negative correlation between miR-181a and miR-27b and the expression of antioxidant-related genes and the GC survival under heat stress. In contrast to the previously discussed miRNAs, miR-708 exhibited the opposite pattern of expression, with a significant upregulation in GCs cultured under heat stress conditions (39.5 °C and 40.5 °C) compared to the control. The expression of miR-708 was correlated with oxidative stress induction in mouse hippocampal neurons [52] and involved in the regulatory network of the Nrf2-mediated antioxidant system in bovine GCs [32]. In this study, we didn’t notice a negative correlation between miR-708 and antioxidant-related genes, including *NRF2*. This could indicate that a cooperatively active mechanism of several miRNAs is required to regulate the expression of antioxidant system genes under heat stress conditions.

## 5. Conclusions

In this study, we demonstrated some physiological and molecular alterations in buffalo GCs induced by heat stress conditions during the last 24 h of in vitro culture (7 days). At 40.5 °C, GCs exhibited a functional persistence compared to the control and other heat-treated groups. This response could be partly through the regulation mechanisms of antioxidant, metabolic, and stress-related genes and miRNAs. However, further studies are required to figure out the molecular regulation of buffalo GCs that underlies the steroidogenesis process. The results of this study will provide insights into ways that tropically adapted breeds may be able to maintain better reproductive function when exposed to heat stress compared to temperate breeds.

## Figures and Tables

**Figure 1 animals-11-00794-f001:**
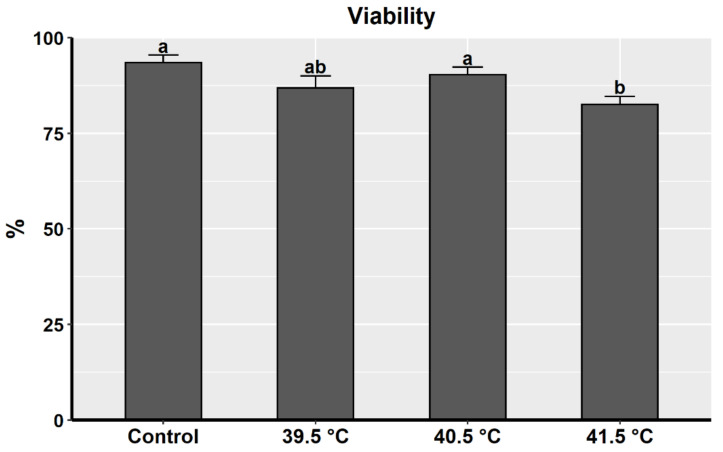
Viability of cultured granulosa cells under heat treatment (39.5, 40.5, and 41.5 °C for the last 24 h of culture, 7 days) and control group (37.5 °C).

**Figure 2 animals-11-00794-f002:**
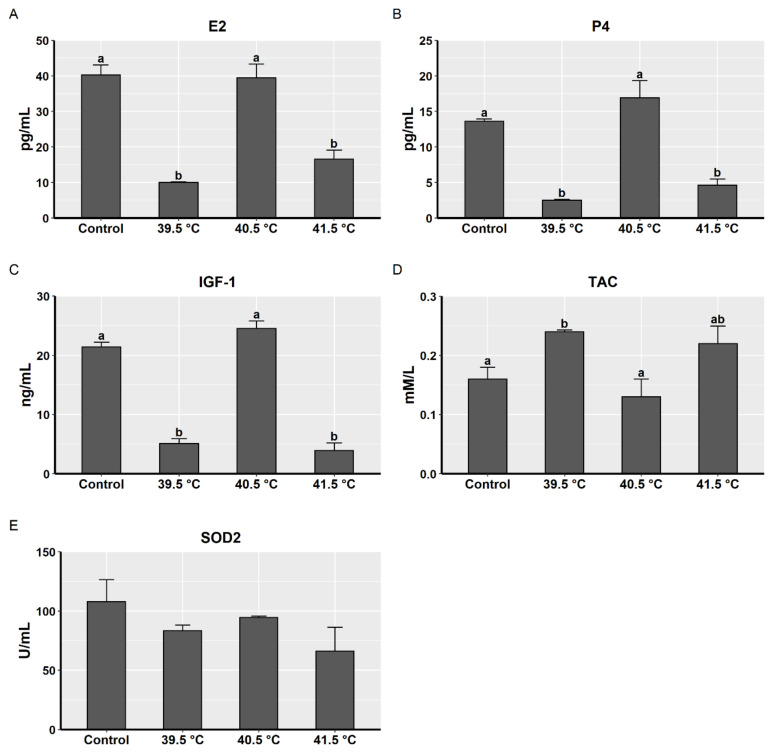
Steroid concentrations and enzymatic activities of buffalo granulosa cells post-heat treatment (39.5 °C, 40.5 °C, or 41.5 °C for the last 24 h of culture, 7 days) in comparison to control group (37 °C); (**A**) E2 (estradiol-17β, pg/mL), (**B**) P4 (progesterone, pg/mL), (**C**) IGF-1 (insulin-like growth factor 1, ng/mL), (**D**) TAC (total antioxidant capacity, mM/L), and (**E**) SOD2 (superoxide dismutase 2, U/mL); data are presented as mean ± SEM; bars with different letters indicate differences (*p* < 0.05); and data from three biological replicates.

**Figure 3 animals-11-00794-f003:**
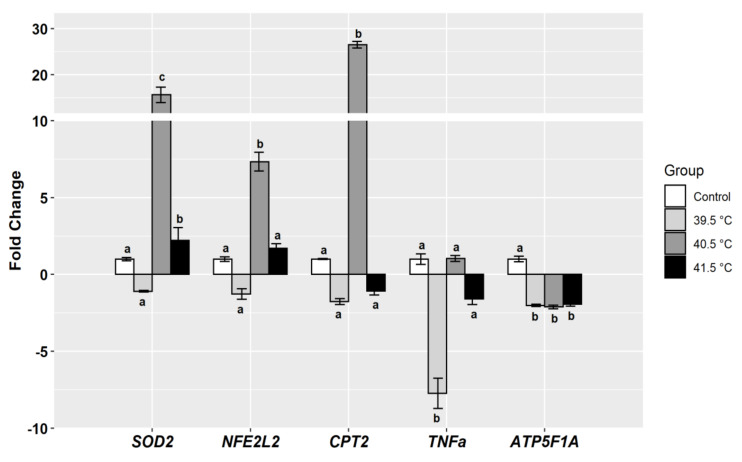
Fold change of some targeted genes post-heat treatment of buffalo granulosa cells (39.5 °C, 40.5 °C, or 41.5 °C for the last 24 h of culture, 7 days) in comparison to control group (37 °C); *GAPDH* was used to normalize the abundances of mRNA transcripts for targets genes; Data are presented as mean ± SEM; bars with different letters indicate significant differences (*p* < 0.05); and data from three biological replicates.

**Figure 4 animals-11-00794-f004:**
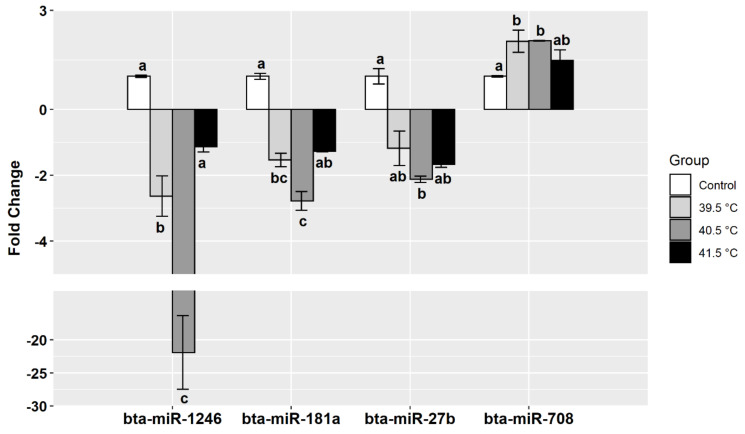
Expression analysis of selected miRNAs in buffalo granulosa cells post-heat treatment (39.5 °C, 40.5 °C, or 41.5 °C for the last 24 h of culture) compared to the control group (37 °C). Bars with different letters indicate significant differences between groups (*p* < 0.05) and data from three biological replicates.

**Table 1 animals-11-00794-t001:** List of oligonucleotide primers used for qRT-PCR reactions.

Gene Name	Accession Number	Primer Sequence	Fragment Size (bp)
*ATP5F1A*	NM_174684.2	F: 5′-CTCTTGAGTCGTGGTGTGCG-3′ R: 5′-CCTGATGTTGGCTGATAACGTG-3′	184
*CPT2*	NM_001045889	F: 5′-CCGAGTATAATGACCAGCTC-3′ R: 5′-GCGTATGAATCTCTTGAAGG-3′	152
*NFE2L2*	NM_001011678	F: 5′-TAAAACAGCAGTGGCTACCT-3′ R: 5′-GAGACATTCCCGTTTGTAGA-3′	159
*SOD2*	NM_201527	F: 5′-GTGATCAACTGGGAGAATGT-3′ R: 5′-AAGCCACACTCAGAAACACT-3′	163
*TNFα*	AF011927	F: 5′-GTGAAGTCGCTCAGTCGTGC-3′ R: 5′-TCTACAAGGCGGGAGACCTG-3′	170
*GAPDH*	NM_001034034.2	F: 5′- AGGTCGGAGTGAACGGATTC -3′ R: 5′- GGAAGATGGTGATGGCCTTT -3′	219

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
