# Peer review of "Adaptive and Biological Responses of Buffalo Granulosa Cells Exposed to Heat Stress under In Vitro Condition"

_animals, 2021, doi:10.3390/ani11030794_

Round 1
Reviewer 1 Report
In this manuscript titled “Adaptive and biological responses of buffalo granulosa cells exposed to heat stress under in vitro condition” the authors tested if buffalo granulosa cell’s viability, physiological and molecular responses under in vitro heat stress. They observed that buffalo GCs exhibited different adaptive responses, to the different heat stress conditions. The integration mechanism between the molecular and secretory actions of the GCs cultured at 40.5 °C might provide possible insights into the biological mechanism through which buffalo GCs react to heat stress.
Although the concept is interesting, the study has a number of major flaws.
1. Study design: This is a catastrophic flaw. They have not even mentioned how many animals they have used or the age of the animals. It appears that they have pooled, if pooled how can they call it biological replicates?
2. Another major issue is that what they claim ‘n’ is not the number of animal used but number of culture replicate, this is not acceptable for this kind of primary cell culture based study. They should have collected ovaries and each animal should be individually cultured and should be counted for ‘n’.
3. Another devastating mistake is the use of HEPES media in CO2 incubator. What was the rationale? This suggest that the authors lack elementary basics of a buffering system. HEPES is used as a bicarbonate alternative. It is a zwitterionic buffer and does not need CO2. They should use physiologically relevant bicarbonate buffer system in a CO2 incubator for long term culture. Using HEPES buffer in CO2 system will for several days will severely affect the buffering system bringing the validity of any observation in that system.
4. Observations based solely on mRNA expressions are biologically less relevant, they should have protein data also.
5. How do the author know that they have collected only granulosa cells, have they tested them with any markers? How did they purify them? What was the percentage of granulosa cell purity?
6. How did they get rid of blood cells?
7. Introduction: First line, what are they talking about? What do they mean by “Their”?
8. In simple summary, they start with global warming, I don’t see any relevance to their study neither have they justified anything in the discussion. In any tropical country summer temperature can easily reach above 40deg C. However, do they have any evidence that the core body temperature of buffaloes reach this high? If so, they need to support their rationale with literature.
Reviewer 2 Report
Dear authors,
This study analyses viability and some molecules, as hormones, growth factors, expression of different genes and miRNAs in the buffalo granulosa cells. The manuscript is well structured and written. However, some major revisions are necessary:
- Lines 20, 28, and others: “in vitro” must be written in italics.
- Line 89: the authors should indicate how many animals have been used, how many ovaries and how many oocytes per animal.
- Line 131: According to the manufacturer, sensitivity for P4 ELISA kit is 154.62 pg/ml, and for E2 is 50 pg/ml. The authors should correct this data and include units.
- Line 131: The manufacturer Sino Gene does not have an ELISA kit for detection of IGF-1 in bovines. It only has for rat, human and chicken. Please indicate which one you have used for the study
- Line 141: the authors should explain exactly how have the groups done.
- Line 147: The authors must indicate the degree of purity obtained and from what value the samples were not analyzed.
- Lines 149 and 177: which concentration of RNA per sample?
- Line 167: the authors must explain why they have used this housekeeping and not another, given that there is a large bibliography showing that a specific housekeeping should be used for a species and for a specific sample. Is there any evidence that this housekeeping is adequate for the type of samples and for the species of the study?
- Line 204: Before performing an ANOVA parametric analysis, the authors should check the normality and homoscedasticity of the data. In most cases, gene expression data do not follow a normal distribution and a mathematical transformation (usually logarithmic) must be applied to normalize them.
- Figure 1: Viability is similar at 39.5ºC and 41.5ºC, could the authors explain these results?
- Line 225: IGF-1 is not a hormone. Please, correct this sentence.
- Line 233: write “the” in capital letters.
- Figure 2: The results are similar with 39.5ºC and 41.5ºC, could the authors explain these results? Why at 39.5ºC there are different results from those obtained at 40.5ºC and, however, similar to those obtained at 41.5ºC where the thermal stress should be greater? What is the possible explanation for these results?
- Lines 248-252: If there are no statistically significant differences, the authors should not speak of a tendency.
- Lines 301-303: This study was carried out by assessing serum oestradiol levels in animals in vivo. In this work, the values in cultured cells have been analyzed, so the results are not comparable. It is true that IGF-1 acts by regulating hormonal secretion, but the authors should include some reference where this is demonstrated in culture and, more specifically, in bovines.
Reviewer 3 Report
The authors studied the response of buffalo follicular granulosa cells (GCs) to different heat stress conditions to identify possible adaptive molecular mechanisms to increased environmental temperatures. For this, isolated GCs were exposed to 39.5. 40.5 and 41.5°C and to 37.5°C as control for 24h after a 7 days culture period.
As readout, cell viability and E2, P4 and IGF-1 concentrations in spent media were determined. In addition, total antioxidant capacity (TAC) and superoxide dismutase enzyme (SOD) were determined in spent media and the abundance of stress related transcripts SOD2, NFE2L2, CPT2, ATP5F1A and TNFa was measured by qRT-PCR. The abundance of selected miRNAs miR-1246, miR-181a, miR-27b miR-708 was determined by droplet digital PCR (ddPCR).
From their data the authors conclude that buffalo GCs exhibited a functional persistence compared to the control and other heat-treated groups at 40.5°C and that molecular and secretory actions of the GCs cultured at 40.5 °C might provide possible insights into the biological mechanism through which buffalo GCs react to heat stress.
Basically, the study provides interesting data on possible compensatory mechanism in buffalo GCs dealing with heat stress. The manuscript is clearly written and the experiments seem properly done and documented in the Figures.
However, the data are partly unsatisfactorily discussed. In addition, it is also a pity that the authors did not determine expression of key transcripts of GC functionality as for example STAR, CYP19A1, HSD3B1 or FSHR and LHCGR. Also data on HIF1 expression and on apoptosis regulating genes (e.g. BCL2, Caspases, BAX) would have been quite interesting under the heat stress conditions. In any case, the authors should better justify the selection of the genes studied. If the authors have these or part of these additional data these should be included in the present manuscript to allow a more comprehensive picture of buffalo GC response to heat conditions.
The discussion does also not provide sufficient clues to understand the observed responses of the cells. The authors discuss mechanism that might be responsible to maintain GC functionality under increased temperature. However, it is still unclear, why the cells do not show any kind of a dose response curve. Instead, the data suggest an up and down under increasing temperatures. The decrease of hormone production at 41.5°C is not discussed and more importantly, the authors do not provide a satisfactory explanation on the obviously different reactions at 39.5 compared to 40.5°C. What do the authors mean with “compromised GCs functionality” in lines 337/338?
Additional data (see above) might have helped to find better explanations and make the study more interesting.
Reviewer 4 Report
Dear editor
The work presented is clear, understandable, and makes interesting contributions to the effects of heat stress on granulosa cell performance in vitro. The experiments are well described and analyze different aspects of cell behavior in vitro. The statistical analysis, the associated presentation of the results is consistent. In the discussion, the results are compared with those obtained in cattle, highlighting the adaptive differences between the two species.
Finally, I consider that it would have been interesting to compare buffaloes with bovines in the same experiment.
Round 2
Reviewer 2 Report
Dear authors,
Thanks for the clarifications and corrections. The manuscript has improved remarkably. However, some indications that you offer in your cover letter should be included in the manuscript. For example, the normality and homoscedasticity analyzes of the data in the statistical analysis, and the references that support the use of GADPH as housekeeping.
Author Response
We would like to thank the reviewer for her/his comments. In line 173, we added the references that support the use of GAPDH as a housekeeping gene and in the statistical analysis section, we included sentences for the normality and homoscedasticity analyzes (Lines 196-199). Please find the changes in the updated version of the manuscript. We checked the other comments from the previous revision and all indications and responses were included in the manuscript accordingly.
Thank you.